# Towards a One Health Assessment of Artisanal and Informal Mining in Benue State, Nigeria †

Samuel N. Paul [1] , Chiara Frazzoli [2] and Orish E. Orisakwe [1,3,*]

1   World Bank African Centre of Excellence for Public Health and Toxicological Research (ACE-PUTOR), University of Port Harcourt, PMB, Choba, Port Harcourt 5323, Nigeria
2   Department of Cardiovascular and Endocrine-Metabolic Diseases and Ageing, Istituto Superiore di Sanità, Via Giano della Bella 34, 00162 Rome, Italy
3   Advanced Research Centre, European University of Lefke, Lefke, Northern Cyprus TR-10, Mersin 99010, Turkey
*   Correspondence: orishebere@gmail.com
†   Presented at the 2nd International One Health Conference, Barcelona, Spain, 19–20 October 2023.

**Abstract:** This study evaluated the eco-health risk associated with exposure to lead (Pb) in mining sites in Benue State, Nigeria. Lead contamination was assessed in 48 water samples and 40 human blood samples. An Atomic Absorption Spectrophotometer was used for Pb analysis and ELISA kits for tumor necrosis factor-alpha (TNF-α) and kidney injury molecule-1 (KIM-1) analysis. A correlation was found between the blood Pb level (BLL) and the upregulation of KIM-1. The BLL of females was slightly higher than males, resulting in a higher inflammatory response through increased TNF-α levels. An increased inflammatory response due to chronic Pb exposure was observed with age. Miners and farmers around the mining sites recorded higher TNF-α levels compared to businesspeople, thus suggesting direct exposure to other mining-associated contaminants. Artisanal and informal mining impact environmental health and the Pb body burden.

**Keywords:** water; eco-health; blood lead level; toxicology; One Health

## 1. Introduction

Eco-health is a field of research, education, and practice that adopts eco-systems approaches to promote the health of people, animals, and the environment. There is increasing global concern about the health effects of heavy metals arising from various anthropogenic activities, especially mining. Mining activities in low-income countries are often carried out at an artisanal level using a variety of extraction methods, with human health and environmental consequences [1]. Mining has long been associated with the release of a range of toxic metals including lead (Pb), found at high levels in land surrounding mine dumps and far afield. Research has documented high levels of Pb in water sources associated with solid mineral mining operations in different world regions [1–4]. Chronic human exposure to Pb in water can lead to accumulation in human blood and elevated blood lead levels (BLLs). Human BLLs are associated with different health implications including oxidative stress, inflammation, apoptosis, renal diseases, and child development issues, among several other health challenges [5]. A positive correlation has been proven between the total Pb concentrations in vegetables/water and children's cognitive function in Nigeria [6].

Mining has been a key pillar of economic development and has left a legacy of major environmental contamination, with the poorest experiencing the highest burden of exposure. Since 1976, Western countries have taken very deliberate steps towards reducing BLLs; these efforts have yielded great results with a progressive decline in BLLs [7,8]. In Nigeria, BLLs are still significantly high especially among exposed workers. Indiscriminate artisanal and small-scale artisanal mining activities cause anthropogenic heavy metal

environmental pollution and biodiversity loss in Nigeria [9,10]. Neurotoxic effects and related neuropathological lesions due to environmental pollution have been observed in the animal population around an artisanal mining site in Zamfara, north-western Nigeria [11]. Some initiatives to introduce safer mining practices in selected communities have been attempted in low-resource small-scale mining communities, e.g., for the reduction in Pb exposure among artisanal small-scale miners and mitigation of off-site contamination and take-home exposure by reducing the dust contamination on clothing and body surfaces [12].

While research is ongoing on bacteria as candidates for the bioremediation of heavy metals in mining sites [13], data on the exposure pathways, e.g., the ingestion of edible plants, dermal contact, and the inhalation of soil particles, will strengthen the scientific evidence in the territory [14]. So far, Pb in soils, floor dust, crops, and especially water seem to be major exposure pathways for blood Pb in the community exposed to mining [15].

The vastness of the informal mining sector and the majority of related work activities across Nigeria remain poorly documented [16]. It is therefore essential to contribute to the national survey and generate data and knowledge to help improve conditions and formulate policies and programs to promote and ensure decent work conditions. This study provides data on the BLL of the local population around solid mining sites in Logo, Benue state, Nigeria, and assesses the health risk of the exposed population and the relevant major pathways.

## 2. Methodology

This study is a comparative cross-sectional study on water sources and human blood samples of individuals living or working around the solid mineral mining sites in Logo, Benue state. Phlebotomy was performed by trained medical laboratory scientists to collect human blood from forty (40) individuals into vacuum blood collection tubes. Full consent was given by these individuals, who were aged 18 years and above. Elabscience ELISA kits were used to analyze tumor necrosis factor-alpha (TNF-$\alpha$) and kidney injury molecule-1 (KIM-1) as human disease biomarkers. The concentration of Pb in the samples was determined using Atomic Absorption Spectrophotometry, Varian AA 240 (AAS). The non-carcinogenic human health risk was calculated using indices such as the Chronic Exposure Dose (CDE), Hazard Quotient (HQ), and Hazard Index (HI), while the carcinogenic human health risk was calculated using the models for Cancer Risk (CR) (as seen in [17–21].

Forty-eight (48) surface water samples were collected from water sources (streams and shallow wells) near the Anyiin, Yonov, Tsukwa, and Ayelamo mining sites (Figure 1). Water samples were collected using 2.5 L Amber Glass VWR® TraceClean® sampling bottles. All collected samples were transferred into dry ice-pack coolers before transportation to the laboratory for analysis. The samples were processed immediately after sampling, and the content of Pb was measured by graphite furnace atomic absorption spectrometry (GFAAS) with Zeeman background correction (AA240Z, Varian) and compared with the limit values determined by the WHO and USEPA [22].

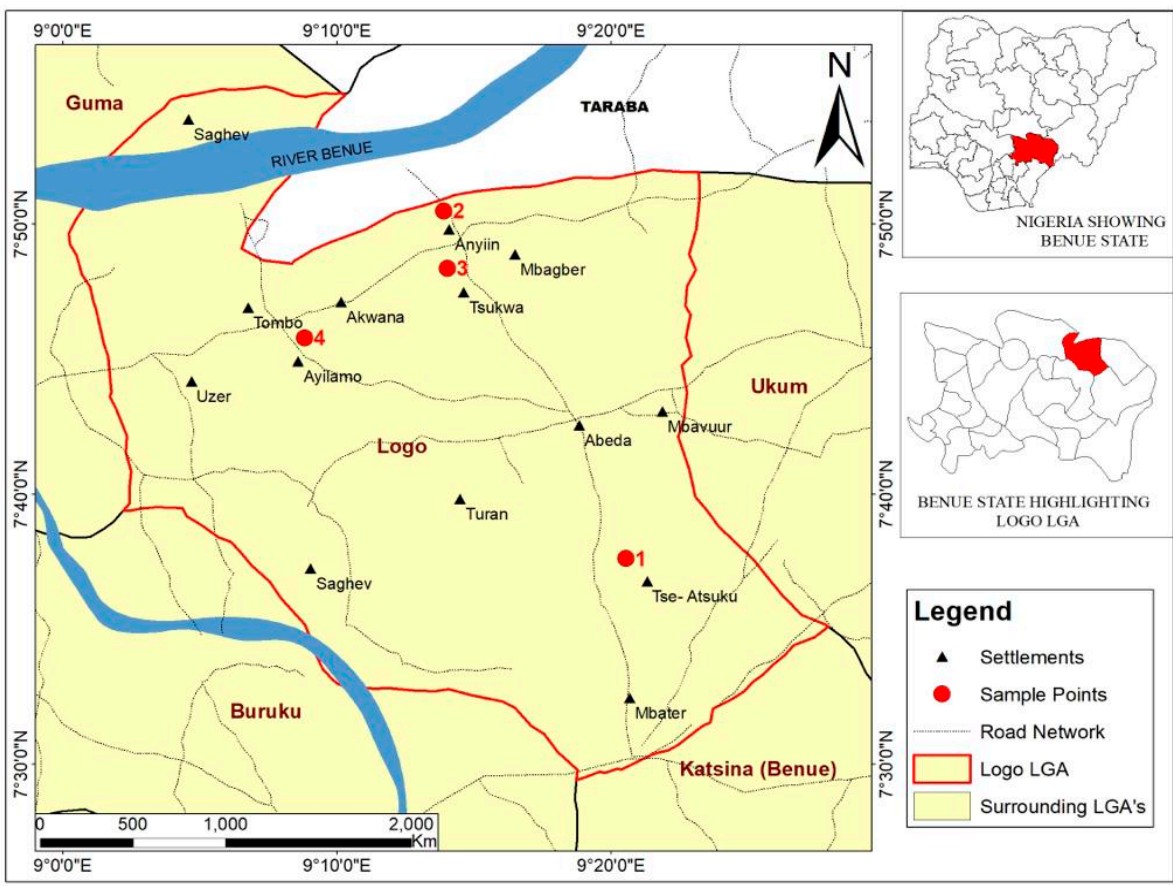

**Figure 1.** Map of Logo local government area showing the various sampling points. LGA = Local Government Area of Benue State, Nigeria.

### 3. Results and Discussion

*3.1. Pb in Water Sources*

Lead contamination of sources of drinking water is a significant public health concern. Exposure to Pb in drinking water significantly affects public health especially in vulnerable groups (infants, young children, and pregnant women) [14,23]. Even at low doses, Pb exposure causes neurological damage and a range of health problems in developing children [24].

The concentration of Pb in drinking water from the mining sites was found to be significantly high, with the following descending order: Anyiin (1.58 ± 0.84 mg/L), Yonov (1.10 ± 0.68 mg/L), Tsukwa (0.57 ± 0.22 mg/L), and Ayelamo (0.48 ± 0.04) (Figure 2). These data are higher than those (0.02–0.9 mg/L) reported by Abara et al. [25] in water from mining sites [26]. These data are also higher than the 0.014, 0.25, 0.695, and 0.082 mg/L in pond water and stream water in abandoned barite mining sites in Cross Rivers state, Nigeria.

The concentrations of Pb in the drinking water sample from the mining sites were all above the standard regulatory limits for Pb in drinking water sources of 0.015 mg/L (15 μ/L) according to the United States Environmental Protection Authority (USEPA) and the limit of 0.010 mg/L (10 μ/L) of the World Health Organization (WHO). The potential risk posed by Pb toxicity in drinking water sources to the population living in the mining areas is high.

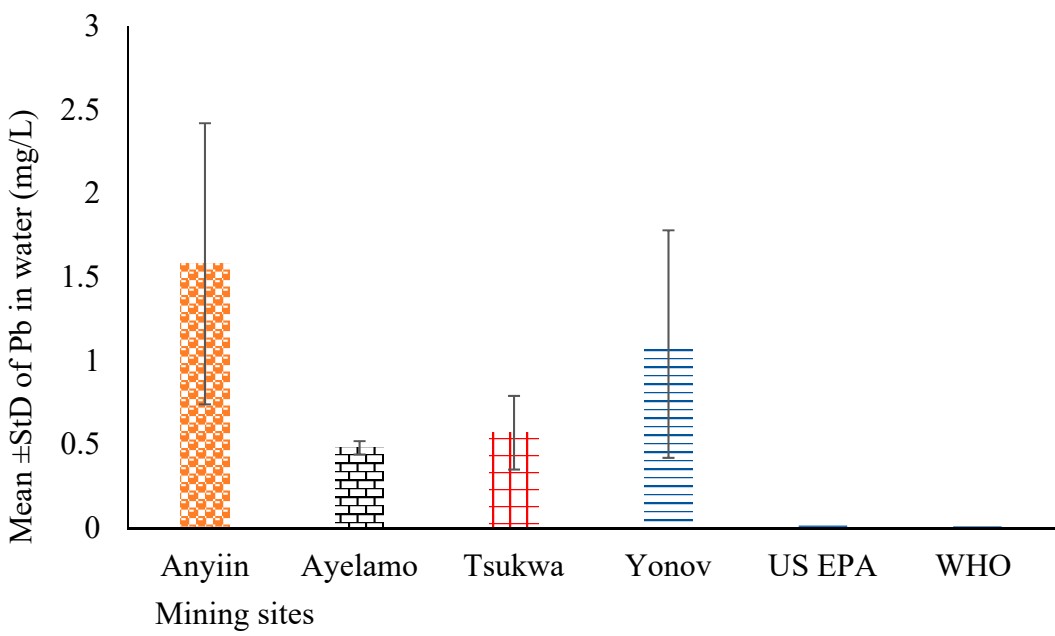

**Figure 2.** Pb (mean ± SD) in drinking water from the mining sites.

### 3.2. Risk Assessment of Pb in Water Samples from the Mining Sites

The CDE through water as presented in Table 1 shows all values are below the standard regulatory value of 1. However, the CDE through ingestion was higher than through derma contact. The HQ of Pb in water from the mining sites in Logo revealed significantly high-risk levels through water ingestion in Tsukwa, Ayelamo, and Yonov for both adults and children. This result is significantly higher than the reports of [27] (2022) for health risk assessment around abandoned barite mining sites in some parts of Benue State. The result of the present study is, however, similar to the reports of Abara et al. [25] (2019) who reported significant HQ (2.687 to 4.742) and HI (2.913 to 5.836) values during a study of heavy metals risk assessment in a dormant mining site in Central Benue Trough. Data on HI due to Pb exposure through water also revealed significantly high values, all above the standard limit value of 1. The carcinogenic risk of human exposure to Pb revealed a moderate-to-low cancer risk for all mining sites in Logo local government area (Logo LGA).

**Table 1.** Chronic Daily Exposure (CDE), Hazard Quotient (HQ), and Hazard Index (HI) of Pb in water.

| | | CDE | | HQ | | HI | CR |
|---|---|---|---|---|---|---|---|
| | | **Derma Contact** | **Ingestion** | **Derma Contact** | **Ingestion** | | |
| Anyiin | Adult | $8.65 \times 10^{-9}$ | $4.77 \times 10^{-8}$ | $2.50 \times 10^{-6}$ | $1.40 \times 10^{-5}$ | $1.61 \times 10^{-5}$ | $7.69 \times 10^{-10}$ |
| | Child | $5.67 \times 10^{-8}$ | $1.52 \times 10^{-7}$ | $1.619 \times 10^{-5}$ | $4.34 \times 10^{-5}$ | $5.96 \times 10^{-5}$ | $3.67 \times 10^{-9}$ |
| Tsukwa | Adult | $3.13 \times 10^{-9}$ | $6.90 \times 10^{-3}$ | $8.95 \times 10^{-7}$ | 1.03 | 1.03 | $3.05 \times 10^{-5}$ |
| | Child | $2.05 \times 10^{-8}$ | $2.19 \times 10^{-2}$ | $5.86 \times 10^{-6}$ | 3.267 | 3.27 | $9.72 \times 10^{-5}$ |
| Yonov | Adult | $2.61 \times 10^{-9}$ | $3.59 \times 10^{-3}$ | $7.44 \times 10^{-7}$ | 0.85 | 0.85 | $2.54 \times 10^{-5}$ |
| | Child | $1.71 \times 10^{-8}$ | $1.14 \times 10^{-2}$ | $4.88 \times 10^{-6}$ | 2.72 | 2.72 | $8.08 \times 10^{-5}$ |
| Yonov | Adult | $6.01 \times 10^{-9}$ | $2.99 \times 10^{-3}$ | $1.71 \times 10^{-6}$ | 1.97 | 1.97 | $5.86 \times 10^{-5}$ |
| | Child | $3.94 \times 10^{-8}$ | $9.51 \times 10^{-3}$ | $1.13 \times 10^{-5}$ | 6.27 | 6.27 | $1.86 \times 10^{-4}$ |

### 3.3. Concentrations of Pb in the Blood of Exposed Individuals around Mining Sites in Logo

Table 2 shows Pb levels in the blood of individuals living around the solid mineral mining sites. The data show that females have a slightly higher mean Pb concentration (5.0 ± 2.0 μg/dL) compared to males (4.0 ± 2.0 μg/dL). Levels of Pb according to age range show little or no variation between the different age ranges. The occupation (male) categories reveal that miners have a slightly lower mean Pb concentration (4.0 ± 2.0)

compared to individuals in business (5.0 ± 2.0) and farming (5.0 ± 2.0). This suggests that gender might moderately contribute to susceptibility to Pb toxicity in the mining sites, and that the environmental contamination exposes the whole population and not only those occupationally exposed.

**Table 2.** Pb, KIM-1, and TNF-α in human blood from the mining sites.

| Group | Parameter | Pb (μg/dL) * | KIM-1 (ng/mL) | TNF-α (pg/m) |
|---|---|---|---|---|
| Gender | Male | 4.0 ± 2.0 | 25.29 ± 13.15 | 7.00 ± 4.72 |
| | Female | 5.0 ± 2.0 | 31.15 ± 16.57 | 10.38 ± 7.43 |
| Age Range | 18 to 29 yrs | 5.0 ± 2.0 | 28.79 ± 15.52 | 8.03 ± 4.35 |
| | 30 to 41 yrs | 5.0 ± 2.0 | 24.45 ± 11.48 | 8.92 ± 8.30 |
| | 42 to 53 yrs | 5.0 ± 2.0 | 38.49 ± 20.68 b | 9.51 ± 4.75 |
| | 54 to 64 yrs | 5.0 ± 4.0 | 31.93 ± 19.41 | 11.10 ± 6.86 |
| Occupation | Mining | 4.0 ± 2.0 | 30.24 ± 17.06 | 8.86 ± 5.23 |
| | Business | 5.0 ± 2.0 | 21.89 ± 8.51 | 5.83 ± 2.39 |
| | Farming | 5.0 ± 2.0 | 29.98 ± 15.81 | 10.22 ± 8.29 |

* Data expressed as mean ± SD.

The result of the present study is slightly lower than the Pb in the blood of electronic technicians in Port Harcourt as reported by Ekine et al. [28]. Ekine et al. reported mean Pb levels of 0.64218 ± 0.9245 (mg/L). In their systematic review, Bede-Ojimadu et al. [29] (2018) reported mean Pb levels in the blood of women of child-bearing age in sub-Saharan Africa as 0.83 to 99 μg/dL.

These levels of Pb in the population around solid mineral mining areas of Logo LGA present a significantly high risk for health. Noticeably, while the US Centers for Disease Control and Prevention (CDC) recommend 5 μg/dL as the standard regulatory limit for Pb levels in human blood, the WHO stipulated that there should not be any acceptable levels for Pb in human blood.

The result of the current study highlights a correlation between BLL and the upregulation of KIM-1 levels. A high BLL results in the accumulation of Pb in proximal tubular cells. This, in turn, leads to the overproduction of reactive oxygen species (ROS) leading to oxidative stress, the subsequent damage of cellular structures, and the excessive expression of proinflammatory cytokines and chemokines creating chronic inflammation. In response to such kidney injury, there is an upward regulation of KIM-1.

Table 2 reveals a clear relationship between a higher BLL and KIM-1 levels as seen among females, miners, and adults between the ages of 42 and 53. This result may be a clear indication of a potential link between Pb exposure and kidney injury. Thus, people exposed to high Pb levels from the mining sites may be at significant risk of Pb-induced nephrotoxicity and possible Pb-associated oxidative stress and inflammation indicating proximal tubular cell injury and the subsequent upregulation of KIM-1.

The result of the present study shows that the BLL of females was slightly higher than males, causing a higher inflammatory response through increased TNF-α levels. The result may suggest that females may be more susceptible to Pb and its associated inflammatory impacts than males.

The Pb blood levels among the different age groups were constant; however, the TNF-α levels increased according to age. The levels in older individuals (54 to 64 years) were significantly higher when compared to the other age groups. This may be an indication of increased inflammatory response due to chronic Pb exposure, as well as weakened immune response and age-related health problems.

The results also show that farmers around the mining sites recorded high TNF-α levels compared to miners and businesspeople. Interestingly, the BLLs of miners were lower than the businesspeople and the farmers. However, miners revealed higher TNF-α levels compared to businesspeople. This may be a result of the impact of the direct exposure to different mining-associated contaminants.

### 4. Conclusions

While the informal sector plays a significant role in the region's economy, policymakers and the scientific community should better understand informal-sector work conditions and catalyze dialogue. Mining activities affect public health both in terms of high BLLs due to the increased Pb concentration in water and the non-carcinogenic health risk. The results indicate possible inflammation and nephrotoxicity. Our results corroborate the need for greater emphasis on the precautionary principle in environmental health and for health impact assessments to inform decisions on (i) planning, especially with regard to the location of human settlements relative to major polluting initiatives, such as mining; (ii) safer mining practices; and (iii) bioremediation. These findings highlight the urgent need for interventions and regulatory strategies to properly manage mining activities so that the health of communities living in the vicinity of a mine is not compromised nor is the environment. The protection of health and well-being of the exposed population and the environment should be a top priority.

**Author Contributions:** S.N.P.: Data acquisition and bench work; C.F.: Drafting of manuscript; O.E.O.: Manuscript writing and conceptualization. All authors have read and agreed to the published version of the manuscript.

**Funding:** This research received no external funding.

**Institutional Review Board Statement:** Study protocol was approved by the University of Port Harcourt, Nigeria with the Ethical approval number: UPH/CERWMAD/REC/MM77/002.

**Informed Consent Statement:** Informed consent was obtained from all volunteers.

**Data Availability Statement:** All data are available within this manuscript.

**Acknowledgments:** Authors acknowledge the generous donation of the ELISA kits by Istituto Superiore di Sanità, Italy.

**Conflicts of Interest:** The authors declare that there are no competing interests.

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
