# Peer review of "Towards a One Health Assessment of Artisanal and Informal Mining in Benue State, Nigeria†"

_msf_

Round 1
Reviewer 1 Report
Comments and Suggestions for Authors
The study provides interesting contributions to the one health field. However, significant changes are needed prior to publication.
1. The English language needs revision. Please carefully revise the abstract, some sentences are difficult to read and grammar needs revisions (e.g., "resulting to higher inflammatory response" needs to be corrected with "resulting in higher inflammatory response"). Please also make sure that all acronyms are explained both in abstract and text (e.g., BLL).
2. Introduction: The section is very short. The authors should clearly explain the gaps, the specific problem(s) that they intend to address, and how they aim to do so.
3. Methodology: This section is very difficult to read and confusing. For example: "Samples were sampled following standard sample collection procedure." This sentence is too repetitive and not informative.
4. Do results start at page 2? It is unclear where methodology ends and where results start. However, the methodology section per se is very poor and needs to be substantially revise to provide a good basis to understand and read the results.
5. The authors do not draw on the implications of the study. What are the contributions to research and practice? What are some new findings and implications compared to existing literature?
6. The conclusion does not draw on the paper's results.
Comments on the Quality of English LanguageThe English language needs revisions throughout the entire manuscript. Some sentences are very repetitive and difficult to read. Further comments are provided above.
Author Response
Manuscript ID
msf-2677976
Type
Proceeding Paper
Title
A pilot One Health assessment of artisanal and informal mining in Benue State, Nigeria.
Authors
Samuel N Paul ,
Chiara. Frazzoli
, ORISH EBERE ORISAKWE *
Comments and Suggestions for Authors
The study provides interesting contributions to the one health field. However, significant changes are needed prior to publication.
- The English language needs revision. Please carefully revise the abstract, some sentences are difficult to read and grammar needs revisions (e.g., "resulting to higher inflammatory response" needs to be corrected with "resulting in higher inflammatory response"). Please also make sure that all acronyms are explained both in abstract and text (e.g., BLL).
RESPONSE: Abstract has been recast for clarity. Title has been tweaked too.
- Introduction: The section is very short. The authors should clearly explain the gaps, the specific problem(s) that they intend to address, and how they aim to do so.
- RESPONSE: INTRODUCTION has been expanded for clarity.
- Methodology: This section is very difficult to read and confusing. For example: "Samples were sampled following standard sample collection procedure." This sentence is too repetitive and not informative.
RESPONSE: Methodology has been recast for clarity.
- Do results start at page 2? It is unclear where methodology ends and where results start. However, the methodology section per se is very poor and needs to be substantially revise to provide a good basis to understand and read the results.
- RESPONSE: Methodology is now clearly separated from RESULT.
- The authors do not draw on the implications of the study. What are the contributions to research and practice? What are some new findings and implications compared to existing literature?
RESPONSE: DISCUSSION has also been expanded in a seamless manner.
- The conclusion does not draw on the paper's results.
- RESPONSE: CONCLUSION has been recast for clarity.

Round 2
Reviewer 1 Report
Comments and Suggestions for Authors
The quality of the manuscript has significantly improved.